# *BdNub* Is Essential for Maintaining gut Immunity and Microbiome Homeostasis in *Bactrocera dorsalis*

**DOI:** 10.3390/insects14020178

**Published:** 2023-02-10

**Authors:** Jian Gu, Ping Zhang, Zhichao Yao, Xiaoxue Li, Hongyu Zhang

**Affiliations:** National Key Laboratory for Germplasm Innovation and Utilization for Fruit and Vegetable Horticultural Crops, Hubei Hongshan Laboratory, Institute of Urban and Horticultural Entomology, College of Plant Science and Technology, Huazhong Agricultural University, Wuhan 430070, China

**Keywords:** *Nub*, *Bactrocera dorsalis*, the antibacterial peptide, gut immunity, gut microbes, IMD pathway

## Abstract

**Simple Summary:**

The innate immune system of insects can recognize various pathogens that invade insects and make rapid immune responses. However, excessive immune activation is detrimental to the survival of insects. *Nub* gene of the OCT/POU family plays an important role in regulating the intestinal IMD pathway. In this study, an important horticultural pest, *Bactrocera dorsalis,* was adopted to study its high adaptability in complex habitats. Through NCBI database analysis, we found that the *BdNub* gene of *B. dorsalis* produced two transcription isoforms, *BdNubX1* and *BdNubX2*. After Gram-negative bacterium *Escherichia coli* with system infection, the immunoeffector genes of Imd signaling pathway, antimicrobial peptides Diptcin (Dpt), Cecropin (Cec), AttcinA (AttA), AttcinB (AttB) and AttcinC (AttC) were significantly up-regulated. The expression levels of antimicrobial peptide genes *Dpt*, *Cec*, *AttA*, *AttB,* and *AttC* were significantly up-regulated at 6 h and 9 h after intestinal infection with the Gram-negative bacterium *Providencia rettgeri*. RNAi showed that the silencing of the *BdNubX1* and *BdNubX2* genes could make the gut more sensitive to *Providencia rettgeri* infection, reduce the survival rate significantly, and cause changes in the gut microbiota’s structure. These results suggest that the maintenance of immune balance plays an important role in *B. dorsalis* high invasiveness.

**Abstract:**

Insects face immune challenges posed by invading and indigenous bacteria. They rely on the immune system to clear these microorganisms. However, the immune response can be harmful to the host. Therefore, fine-tuning the immune response to maintain tissue homeostasis is of great importance to the survival of insects. The *Nub* gene of the OCT/POU family regulates the intestinal IMD pathway. However, the role of the *Nub* gene in regulating host microbiota remains unstudied. Here, a combination of bioinformatic tools, RNA interference, and qPCR methods were adopted to study *BdNub* gene function in *Bactrocera dorsalis* gut immune system. It’s found that *BdNubX1*, *BdNubX2*, and antimicrobial peptides (AMPs), including *Diptcin* (*Dpt*), *Cecropin* (*Cec*), *AttcinA* (*Att A*), *AttcinB* (*Att B*) and *AttcinC* (*Att C*) are significantly up-regulated in Tephritidae fruit fly *Bactrocera dorsalis* after gut infection. Silencing *BdNubX1* leads to down-regulated AMPs expression, while *BdNubX2* RNAi leads to increased expression of AMPs. These results indicate that *BdNubX1* is a positive regulatory gene of the IMD pathway, while *BdNubX2* negatively regulates IMD pathway activity. Further studies also revealed that *BdNubX1* and *BdNubX2* are associated with gut microbiota composition, possibly through regulation of IMD pathway activity. Our results prove that the *Nub* gene is evolutionarily conserved and participates in maintaining gut microbiota homeostasis.

## 1. Introduction

Insects, composed of over 5 million different species, are the most abundant species on earth and can survive in all kinds of complex environments [1]. These highly diverse habitats also exert tremendous survival pressure on insects. Infection by environmental microorganisms and colonization by indigenous bacteria can be detrimental to insects [2]. During their long-term evolution, insects, like all other animals, developed efficient immune defense systems. Although lacking an adaptive immune response, insects can resist bacteria, fungi, viruses, and nematodes using their innate immunity. The IMD pathway plays a vital role in defense against Gram-negative bacteria. This process is marked by antimicrobial peptides (AMPs) production and phagocytosis of bacteria accomplished by blood cells [3,4,5]. Insects also synthesize and accumulate many immune effectors, which are released into the hemolymph and play a role in the immune response, which is called the humoral response [6].

In insects, humoral immune effectors are mainly AMPs. About 20 inducible antimicrobial peptides have been identified in *Drosophila melanogaster*, showing a broad spectrum of antimicrobial effects [7]. AMPs can act either in a specific or synergistic way [8]. In *Drosophila* and honeybee, *Diptericin*, *Drosocin*, and *Attacin* play essential roles in the defense against Gram-negative bacteria [7,9]. *Defensin* mainly resists Gram-positive bacteria, while *Drosomycin* and *Metchnikowin* are the main active antifungal substances. Cecropin plays an essential role in both anti-bacterial and anti-fungal processes [10], and the constitutively expressed antimicrobial peptide *Andropin* is continuously expressed in male reproductive organs for defense [11].

The *Nub* gene of the POU/OCT family was early discovered in *D. melanogaster* [12,13]. The POU/OCT family regulates key regulators of metabolism, immunity, and cancer [14]. The *Nub* gene encodes two distinct proteins with independent functions, *Nub-PB* and *Nub-PD*. These two proteins have similar expression patterns but perform different functions. The difference between *Nub-PB* and *Nub-PD* proteins’ N-terminal sequence leads to a different regulatory mechanism. *Nub-PD* is a transcriptional repressor of the antimicrobial peptide gene, while *Nub-PB* is a transcriptional activator of the antimicrobial peptide gene in *Drosophila* [14,15]. In *Drosophila*, overexpressing *Nub-PB* results in increased AMPs abundance [15]. On the other side, overexpressing *Nub-PD* results in reduced AMP abundance. Furthermore, co-overexpressing *Nub-PB* and *Nub-PD* does not induce changes in AMPs gene expression [14,15].

Microorganisms are found in many plants, animals, and other organisms [16]. Insects host probably the largest group of commensal bacteria [2]. These bacteria are abundant in the gut, body cavity, and specific cells of insects [2]. They have a wide range of functions, including contributions to host growth and development, nutrient acquisition, and resistance to pathogens [17,18,19,20]. On the other hand, dysregulation of gut microbiota can also be harmful to the host [21]. Therefore, the gut microbiota needs to be tightly controlled. In *Drosophila*, proper gut microbiota composition, density, and localization were altered in IMD-deficient flies, suggesting IMD’s prominent role in gut microbiota control [22]. Maintaining gut microbiota homeostasis and eliminating pathogenic bacteria are essential to the host’s health. In a recent study, compartmentalized IMD pathway receptors *PGRP-LC* and *PGRP-SB*, *PGRP-LB* expression act to eliminate pathogenic bacteria and protect symbiotic bacteria in *Bactrocera dorsalis* [23].

*B. dorsalis* is a major horticultural and agricultural pest. It damages more than 250 kinds of fruits and vegetables, causing substantial economic loss. Its larva lives inside rotten fruits and faces threats posed by bacteria. Therefore, its immune system, especially the gut immune system, must be precisely regulated to ensure its survival [21,24,25]. Since Nub gene is essential for AMPs gene expression regulation, we expect it also plays an indispensable role in *B. doraslis* gut immunity. In this paper, we also aim to decipher *BdNub* function in *B. dorsalis* microbiota homeostasis. Gut microbiota has been proven to be necessary for *B. dorsalis* overall fitness [20,24]. Our findings suggest that *BdNub* could be a novel target for developing pest control strategies targeting gut homeostasis.

## 2. Materials and Methods

### 2.1. Insects Rearing

The experimental insects were collected from Guangdong Province, China using protein bait and maintained in the Institute of Urban and Horticultural Insects, Huazhong Agricultural University, Wuhan, Hubei, China. The photoperiod of the insect-rearing room was 12 h:12 h. The room’s relative humidity was 70–80%, and the temperature was maintained at 28 ± 1 °C. Larvae were raised on larval food (wheat bran 80 g, corn flour 40 g, sucrose 40 g, yeast powder 15 g, water 200 mL). After eclosion, adult flies were moved to 30 cm × 30 cm × 30 cm cages. Adult flies were raised on a sucrose and yeast mix at a ratio of 3:1. 

### 2.2. BdNub Identification 

We blasted the *Drosophila Nub* protein (NCBI REFSEQ: accession NM_001103683.2) sequence against NCBI. *BdNub* BLAST results showed that *BdNub* has two different transcripts. The online analysis tool SPLIGN was used to identify *BdNub* gene introns and exons. The Neighbor-Joining phylogenetic tree was built using MEGA7. DNAMAN was used to perform amino acid homology analysis. We used SnapGene to analyze nucleotide sequence homology. The online tool SMART was used to predict and analyze the conserved protein domains. The protein secondary structure was analyzed using the online tool SOPMA to submit amino acid sequences to the SOPMA working page. The results show an alpha helix (blue), a beta turn (green), a random coil (yellow), and an extended strand (red). SWISS-MODEL was deployed to predict protein structures, input the target amino acid sequence, and build a model. A simple way to evaluate the quality of a model is to look at the GMQE value (Global Model Quality Estimate), which is between 0 and 1. The closer to 1, the better the quality of the model.

### 2.3. BdNub Spatial and Temporal Expression Profiles

For spatial analysis, adult flies were dissected. Tissue samples were collected from the head, gut, Malpighian tubules, ovary, testis, and fat body. The samples were stored at −80 °C for further use. For temporal analysis, whole eggs or insects were collected from different developmental stages, including eggs, first instar larvae, second instar larvae, third instar larvae, one-day-old pupa, nine-day-old pupa, one-day-old adult flies (male and female apart), and 14-day-old mature adult flies. Samples from different developmental stages were rinsed once in 75% alcohol, followed by two rinses in PBS.

### 2.4. RNA Extraction and cDNA Synthesis

For spatial expression profiles, total RNA was isolated from 30 different tissue samples per biological replicate; three biological replicates were conducted. For temporal expression profiles, total RNA was isolated from 20 different stage of development samples per biological replicate; three biological replicates were conducted. Samples were placed into a 1.5 mL RNase-free centrifuge tube containing ground beads and homogenized (Jinxin, Shanghai, China). Total RNA was extracted using Trizol (TaKaRa, Otsu, Shiga, Japan) following the manufacturer’s instructions. RNA quality and concentration were determined by electrophoresis (Liuyi Biotechnology, Beijing, China) and a Nanodrop spectrophotometer (Thermo Fisher Scientific Inc., Waltham, MA, USA). RNA was stored at −80 °C for later use. The first-strand complementary DNA (cDNA) of each pool was synthesized from 1 μg of total RNA using the PrimeScriptTM RT reagent kit (Takara, Otsu, Shiga, Japan) with a gDNA eraser to remove residual DNA contamination.

### 2.5. Real-Time PCR

Real-time PCR was performed using the Hieff UNICON^®^ qPCR SYBR Green Master Mix No Rox kit (Bio-Rad, Hercules, CA, USA)on a real-time Bio-Rad CFX96 (Bio-Rad, Hercules, CA, USA) PCR instrument. Real-time PCR was performed using a Bio-Rad CFX Connect system with the following protocol: initial denaturation at 95 °C for 30 s, followed by 45 cycles of 95 °C for 15 s and 60 °C for 30 s. Melting curve analysis was performed at the end of each amplification run to confirm the presence of a single peak with the following protocol: 55 °C for 60 s, followed by 81 cycles starting at 55 °C for 10 s with a 0.5 °C increase each cycle. Real-time PCR results were relatively quantified using the 2^−ΔΔ^_CT_ method [26]. Relative mRNA abundance was normalized using the *RPL32* set as a reference gene. Real-time PCR primers> are listed in Table 1. Each sample was set in a triplicate, and the corresponding blank control was set as required. The total PCR reaction volume was 20 μL, including 10 μL SYBR Green mix, primers 1.6 μL, RNA-free water 6.4 μL, and cDNA 2 μL. The data was analyzed and exported using Graphpad 7.0.

### 2.6. DsRNA Synthesis and RNA Interference

*BdNubX1* and *BdNubX2* specific primers were designed using Prime5 software (Table 2). For *BdNubX1* dsRNA synthesis, the T7 polymerase recognition sequence (GGATCCTAATACGACTCACTATAGGN) was added to the 5′ of the primers. *BdNubX1* dsRNA was synthesized using the T7 Ribomax express RNAi system (Promega, Madison, WI, USA). *Egfp* dsRNA was synthesized as a control. Our initial assessment showed that *BdNubX2* RNAi could not be achieved by dsRNA injection. Therefore, we choose siRNA for *BdNubX2* RNAi. *BdNubX2* siRNA was synthesized using specific primers of *BdNubX2* (RiboBio, Guangzhou, China).

DsRNA integrity and concentration were monitored by 1% agarose gel electrophoresis and a NanoDrop 2000 spectrophotometer (Thermo Fisher Scientific Inc., Waltham, MA, USA). Microinjection was performed using the FemtoJet 5247 micromanipulation system (Microinjector for cell biology, FemtoJet 5247, Hamburg, Germany) with a Pi of 300 hpa and a Ti of 0.3 s. One microliter of 2 μg/μL dsRNA was injected into the adult flies’ abdomen [23]. For ds-*egfp*, ds-*BdNubX1*, si-*egfp*, and si-*BdNubX2* treatments, we used 100 flies (age: 2 days after emergence) for injection.

### 2.7. Bacterial Infection and Survival Assay

*Escherichia coli* was used for systemic infection. *E. coli* culture was left to grow in 200 mL LB broth for 14 h at 220 rpm at 37 °C. *E. coli* was harvested by centrifuging at 3600 g for 5 min. The final concentration of the *E. coli* solution was adjusted to OD_600_ = 400(~10^11^ cfu/mL). For pricking, we used flies 5 days after RNAi treatment. Briefly, a clean insect needle was first surface sterilized by ethanol and then dipped into the bacterial pellet. Flies were then pricked in the abdomen. For oral infections, we used the gram-negative bacteria *P. rettgeri*. The *P. rettgeri* strain used in this study was isolated from the *B. dorsalis* gut. It could induce a strong immune response through oral infection in adult *B. dorsalis*. The culture method was the same as previously described. The final bacteria concentration for oral infection was OD_600_ = 50(~10^10^ cfu/mL). Adult flies were starved and dehydrated for 24 h without food or water supplies. For infection, flies were then fed an artificial diet with 5% sucrose containing the concentrated microbe solution [23]. The control group was fed only 5% sucrose.

For the survival assay, *BdNubX1* and *BdNubX2* were silenced by injecting corresponding dsRNA and siRNA, and we used 30 flies (age: 2 days after emergence) for injection per biological replicate, and three biological replicates were conducted, respectively. The ds*Egfp*-injected flies were used as the control group. We fed the flies with *P. rettgeri* for 24 h. Next, the infected flies were switched to the normal adult diet. Survival was recorded every day. 

### 2.8. Gut Bacterial DNA Extraction

*BdNubX1* RNAi, *BdNubX2* RNAi, and the control flies (age: 2 days after emergence) were surface sterilized using 75% ethanol and then transferred to PBS (pH = 7.2). DNA was extracted from 30 gut regions of flies (age: *BdNubX1* RNAi; *BdNubX2* RNAi after 3 d, 5 d, and 7 d) per biological replicate; three biological replicates were conducted. According to the manufacturer’s instructions, total DNA was extracted using an E.Z.N.A. Soil DNA kit (Omega, Norcross, GA, USA). 

### 2.9. Quantification of Bacterial Species or Group by qPCR

For bacteria quantification, qPCR was carried out in a 20 mL reaction volume that included 10 mL of SYBR Green Mix (Bio-Rad, Hercules, CA, USA), 200 nM of each primer, and 5 ng of DNA. Real-time PCR was performed using a Bio-Rad CFX Connect system with the following protocol: (1) preincubation at 50 °C for 2 min and 95 °C for 10 min (2) 45 cycles of denaturation at 95 °C for 15 s and annealing at 60 °C for 1 min; and (3) one cycle at 95 °C for 15 s, 53 °C for 15 s, and 95 °C for 15 s. Real-time quantitative PCR (qPCR) was performed and normalized to the host *β-actin* gene (Table 2). In this study, bacterial primers are listed in (Table 3).

### 2.10. Statistics and Analysis

Student’s *t*-test was used for two independent samples to compare the mean values. For multiple comparisons among samples, the Tukey’s test in one-way ANOVA was used, and the significance level was set at *p* < 0.05. An analysis of variance was completed with SPSS 18 software. GraphPad Prism 7.0 and Excel software were used to plot and analyze the experimental data.

## 3. Results

### 3.1. Identification of Nub Gene in B. dorsalis

We found only one *Nub* gene (accession number: NW_011876379) in *B. dorsalis* by searching the NCBI database based on protein sequence homology using BLASTp. *BdNub* has two different transcripts, which we named *BdNubX1* and *BdNubX2*, respectively. *BdNub* consists of 6 exons. *BdNubX1* contains five exons, including transcript-specific exons 1 and 2, which are missing in *BdNubX2*, while *BdNubX2* has four exons, including transcript-specific exon 3 (Figure 1A). *BdNubX1* encodes 833 amino acids, and *BdNubX2* encodes 567 amino acids. The two transcripts share the same 529 amino acids. Sequence alignment showed that *BdNubX1* has 304 transcript-specific amino acids, while *BdNubX2* has 38 transcript-specific amino acids. The *Nub* belongs to the OCT/POU family genes, evolutionarily conserved from arthropods to mammals. We performed the phylogenetic analysis on insects, including *B. dorsalis*, *D. melanogaster*, *Musca domestica*, *Ceratitis capitata*, *Bombyx mori*, and *Aedes aegypti*. The results confirmed that the *Nub* gene is conserved in insects. The results also revealed that despite sequence differences, two *Nub* isoforms clustered in the same branch. *BdNub* was closely related to *Bactrocera capsicum* (Figure 1B).

The *Nub* gene has two POU/OCT family conserved domains: the POU domain and the HOX domain. Although *BdNubX1* and *BdNubX2* have different amino acid sequences, they both have two intact conserved domains. We compared the amino acid sequences of the POU and HOX domains of *BdNubX1* and *BdNubX2* (Figure 1C). The amino acid sequences of *BdNub* POU and HOX are highly similar to the functional domains of *B. capsicum* and *D. melanogaster*, suggesting their functional similarity in vivo. We further use three other algorithms, including SMART, SOPMA, and SWISS, to predict the conserved domain and protein structure of *BdNub*. We also confirmed that *BdNub* is a classical POU/OCT family member (Appendix A).

### 3.2. Temporal and Spatial Expression Patterns of the BdNub Gene

We analyzed the spatial and temporal expression patterns of *BdNub* gene transcripts at different periods and in different tissues. The results showed that *BdNubX1* was highly expressed in 9-day-old pupa (the late-stage pupa) and the freshly emerged adults, and the expression was low in the egg, larva, and sexually mature adult stages (Figure 2A). Similarly, *BdNubX2* was also highly expressed in the late pupal stage and the newly emerged adult flies (Figure 2B). Spatial analysis revealed that *BdNubX1* expression was highest in the midgut, followed by the foregut and testis (Figure 2C). *BdNubX2* showed a slightly different expression pattern. Although it was highly expressed in the foregut and midgut, it was much lower in the testis compared with *BdNubX1* (Figure 2D). These results suggested that two *BdNub* transcript isoforms have similar expression patterns with high abundance in the late pupal stage, newly emerged adult flies, and in the adult gut.

### 3.3. BdNub Does Not Participate in Systemic Infection of the IMD Pathway

Next, we examined *BdNubX1* and *BdNubX2* expression at different time points after *E. coli* systemic infection. *E. coli* is a gram-negative pathogen that can elicit a robust immune response in *B.dorsalis*. The results showed that the *BdNubX1* transcript had no significant increase after systemic infection with *E. coli* (Figure 3A). Similarly, the *BdNubX2* transcript showed no significant change after *E. coli* infection (Figure 3B). These results indicated that *BdNub* did not play a role in the systemic infection of Gram-negative *E. coli*. To strengthen our conclusion, we further detected the main immune effectors of the IMD pathway, including AMPs *Dpt*, *Cec*, *AttA*, *AttB*, and *AttC*. The results showed that all five AMPs were significantly up-regulated at 3 h, 6 h, and 12 h after *E. coli* infection, proving that systemic infection works well in *B. dorsalis* (Figure 3C–E).

### 3.4. BdNub Regulates the Expression of Gut AMPs after Oral Infection

Previous data in our lab suggests that *E. coli* could not induce a strong and consistent gut immune response in *B. dorsalis*. Our screening found that *P. rettgeri* is a potent inducer of the gut immune response. Our results showed that *BdNubX1* was significantly up-regulated at 6 h post oral infection of *P. rettgeri*, while there was no significant up-regulation at other time points (Figure 4A). This suggested that *BdNubX1* is only transiently activated during the gut immune response. *BdNubX2* showed a somewhat different expression pattern. It was significantly up-regulated at 6 h and 9 h after *P. rettgeri* oral infection, suggesting it played a prolonged role in gut immunity (Figure 4B). Next, to reaffirm that *P. rettgeri* could induce a gut immune response, we further examined AMP expression at different time points after infection. The results showed the immune effector AMPs’ expression, including *Dpt*, *Cec*, *AttA*, *AttB*, and *AttC*, increased at 6 h and 9 h after *P. rettgeri* oral infection. The results are as follows: at 6 h after oral infection, all five AMPs genes were significantly up-regulated (Figure 4C), while at 9 h after oral infection, only *Dpt* and *Cec* were significantly up-regulated (Figure 4D). 

Next, we performed RNAi experiments to elucidate the function of the *BdNub* genes. The results showed that *BdNubX1* RNAi down-regulated IMD target AMP genes, including *Dpt*, *Cec*, *AttA*, *AttB*, and *AttC* (Figure 4E,F). It indicated that *BdNubX1* has a positive regulatory function on the IMD pathway. On the contrary, *BdNubX2* RNAi leads to a significant up-regulation of the IMD-regulated AMPs expression (Figure 4G,H), suggesting that *BdNubX2* has a negative regulatory function on the IMD pathway (Figure 4G,H). In order to further explore the function of the *BdNub* gene in gut immunity, we performed gut infection after RNAi of *BdNubX1* and *BdNubX2*, respectively. The results showed that the expression levels of AMPs in *BdNubX1*-silenced flies after gut infection were significantly down-regulated compared with those in control flies after gut infection, and the expression levels of AMPs were similar to those of non-gut infection (Figure 4I). On the contrary, the expression levels of AMPs were significantly up-regulated in the *BdNubX2*-silenced flies after gut infection compared with the control flies after gut infection (Figure 4J). 

Furthermore, to determine whether *BdNubX1* and *BdNubX2* RNAi affect *B. dorsalis* survival, we examined the survival rate of adult flies after *P. rettgeri* oral infection. The results showed that *BdNubX1* RNAi flies fed on *P. rettgeri* died faster than the control group, which was provided with only sucrose. This demonstrated that *P. rettgeri* is indeed a pathogenic bacterium in *B. dorsalis*. On day 20, *BdNubX1* RNAi flies showed a much lower survival rate than the *Egfp* RNAi control group (Figure 4K). Similarly, the results showed that *BdNubX2* RNAi flies fed on *P. rettgeri* died faster than the control group feeding only sucrose. On day 19, *BdNubX2* RNAi flies showed a lower survival rate than the *Egfp* RNAi control group (Figure 4L). 

### 3.5. BdNub Is Necessary to Maintain Gut Microbiota Composition and Structure

The IMD pathway is involved in insect gut microbiota regulation [2,22]. To determine the function of *BdNubX1* and *BdNubX2* isoforms in microbiota regulation, we examined microbiota composition using qRT-PCR in *BdNubX1* and *BdNubX2* RNAi flies. Our results showed that *BdNubX1* RNAi disturbed gut microbiota composition and decreased microbiota abundance (Figure 5A,C,E,G). The bacterial loads of different genera have changed significantly, with strains like *Lactobacillus* and *Enterococcus* significantly down-regulated, and *Pseudomonas* and *Salmonella* significantly increased (Figure 5D,F,H). This result indicated that *BdNubX1* RNAi changed gut microbiota composition and quantity. Similarly, *BdNubX2* RNAi also caused significant changes in gut microbiota. The total intestinal bacteria were down-regulated on day 5 after the dsRNA injection (Figure 5B,I,K,M). Among them, the *Enterobacteriaceae* were significantly down-regulated on days 3 and 5. The abundance of *Pseudomonas*, *Flavobacterium*, and *Salmonella* also changed at different time points after RNAi (Figure 5J,L,N). 

## 4. Discussion

In this study, we report that *BdNub* encodes two isoforms, *BdNubX1* and *BdNubX2*. They are involved in the gut immune response, not systemic immunity, possibly by regulating the gut IMD pathway. Our results also show that *BdNub* is essential for maintaining gut microbiota.

We identified a putative *B. dorsalis Nub* gene, *BdNub*. Similar to *Drosophila*, it also produces two distinct *BdNub* isoforms [15]. Phylogenetic tree analysis shows that the *BdNub* gene is closely related to *B. capsicum,* and the *Nub* gene is highly conserved in insects. This is also confirmed with the protein alignment of conserved domains, POU and HOX [33]. There are five different POU proteins in the *Drosophila* genome [34], regulating embryonic development and differentiation [12], immune function, and tissue homeostasis [14,15]. In addition, pdm1, a POU family gene, acts as proximal-distal growth of the wing, which has a similar function to *Nub* [35]. Moreover, the HOX domain regulates muscle and wing development [36,37], specifying the anterior posterior axis in all bilaterians [38]. So, this indicates the high expression of *NUB* genes in the old pupa stage and the day 1 adult flies.

Spatial and temporal expression pattern analysis showed that *BdNubX1* and *BdNubX2* were mainly expressed in the late-stage pupa, which may be related to Nub gene function in wing development [39,40]. In *Drosophila*, *Nub* mutant flies showed a severe wing size reduction [35,40,41,42], indicating that *BdNub* may also have an indispensable role in *B. dorsalis* wing formation. This might relate to the HOX domain of *BdNub*. Furthermore, we also observed high *BdNub* expression in newly emerged adults and guts. *B. dorsalis* must crawl out of the soil to accomplish eclosion. Thus, newly merged adult flies are exposed to various microorganisms from the soil and environment. High *BdNub* expression at this stage suggested its role in regulating gut immune balance in this process [43].

Our results showed that *E. coli* systemic infection induced the AMP genes’ expression but not *BdNubX1* and *BdNubX2*. It suggests that the *BdNub* gene is not involved in *E. coli*-induced systemic immunity. This result is consistent with their high expression in the gut and low expression in the fat body, the major systemic immune organ. However, immunostaining reveals that *Nub* protein is present in the fat body of *Drosophila* [14]. This suggests *Nub* could be functional in regulating *Drosophila’s* systemic immune response.

In contrast, Gram-negative bacteria *P. rettgeri* oral infection induced a strong immune response and a strong *BdNubX1* and *BdNubX2* up-regulation, suggesting that *BdNub* was involved in the gut immune response. Furthermore, we showed that *BdNubX1* positively regulated gut AMP expression, while *BdNubX2* inhibited AMP expression. The fact that *BdNubX2* is an immunosuppressor of IMD pathway activity is also in line with the *Nub* gene function in the *Drosophila* gut [7,43]. In *Drosophila*, *Nub-PD* RNAi increased AMP gene expression. Similarly, *BdNubX2* RNAi also increased AMP gene expression. Apart from *Nub*, the IMD signaling pathway is regulated by many other factors. For example, the *Pirk* gene encodes the protein binding *PGRP-LC*. It is regulated by the IMD pathway itself. Nevertheless, it establishes a negative feedback loop adjusting IMD pathway activity [44]. *PGRP-SB* and *PGRP-LB* are secreted proteins with an amidase activity that scavenges DAP-type peptidoglycan [6]. They negatively regulate the IMD pathway. A recent study also shows that *PGRP-SB* and *PGRP-LB* are negative regulators of the gut IMD pathway in *B. dorsalis* [23,45,46]. There may also be other unknown factors that regulate the gut IMD pathway activity that have yet to be identified.

In addition, both *BdNubX1* and *BdNubX2* RNAi could make the flies more sensitive to *P. rettgeri* infection. Nevertheless, there was no significant difference in the mortality rate, which was consistent with the results of *Drosophila* [15]. This may be due to the fact that RNAi could not achieve a stable and long-lasting silence effect in *B. dorsalis*. Furthermore, the efficiency of RNAi varies considerably among insects, although RNAi can reach 90% efficacy in *Coleopterans* [47]. *Dipteran* species are not very sensitive to RNAi [48]. Studies using null mutants should be carried out to further elucidate the *Nub* function in gut immunity in the future. Moreover, unlike *Drosophila*, our screening did not find any strong lethal pathogenic bacteria for the gut infection. Although *P. rettgeri* could induce a strong gut immune response, it kills the wild-type flies very slowly, which might be an immune tolerance phenotype rather than an immune resistance phenotype. It indicates that the high adaptability of *B. dorsalis* may be related to its strong immune system.

Since *BdNubX1* and *BdNubX2* transcript isomers play an important role in intestinal immunity, it is plausible that they also participate in controlling the microbiota. Many reports show that changes in immune-related genes will lead to changes in intestinal microbial community structure [21,23]. *BdNubX1* knockdown leads to significantly up-regulated total bacterial abundance, with increased *Pseudomonas* and *Salmonella*. This indicates *BdNubX1* partially regulates gut microbiota composition and abundance through the IMD pathway. The IMD pathway is an important part of the insect gut microbiota regulation mechanism [49]. For example, *PGRP-LB* and *PGRP-SB* have high expression levels in the anterior and middle midgut, which is associated with gut commensal bacteria distribution in *B. dorsalis* [23]. In *Drosophila*, IMD-deficient flies showed a dysregulated gut microbiota and disturbed gut homeostasis [22]. Moreover, we cannot exclude the possibility that *BdNubX1* also interacts with other gut immune mechanisms. For example, *BdDuox* regulates gut microbiota through the production of ROS [21]. Therefore, it is possible that *BdNubX1* might regulate microbiota by affecting ROS production and scavenging.

On the other side, *BdNubX2* knockdown leads to decreased gut microbiota and *Enterobacteriaceae*. It also causes changes in *Pseudomonas*, *Flavobacterium,* and *Salmonella* abundance. In *B. dorsalis*, *Enterobacteriaceae* bacteria account for a large proportion in the gut and are the dominant bacterial group in gut microbiota [50]. In fact, the key ROS production enzyme, *Bdduox* RNAi, induces a similar changes in microbiota composition, with decreased *Enterobacteriaceae* and a rise in secondary microbiota abundance [21]. Several studies have shown that *Enterobacteriaceae* bacteria are beneficial to the host [20,24]. Therefore, decreased *Enterobacteriaceae* in *B. dorsalis* is possibly detrimental to the host. Our results suggest that *BdNubX2* could also be essential to host development and homeostasis through regulating *Enterobacteriaceae* bacteria, which could contribute to the early death of *BdNubX2* RNAi flies. Actually, *Drosophila Caudal* mutants show constitutively activated AMP genes, leading to an increase in *Gluconobacter sp*., causing an early death of the hosts [49]. Altogether, we proved that the *BdNub* gene regulates the IMD pathway to maintain intestinal microbial homeostasis. In conclusion, *BdNub* plays an important role in the regulation of intestinal immunity, decreasing the host’s sensitivity to intestinal opportunistic pathogens, and regulating gut microbiota. 

## Figures and Tables

**Figure 1 insects-14-00178-f001:**
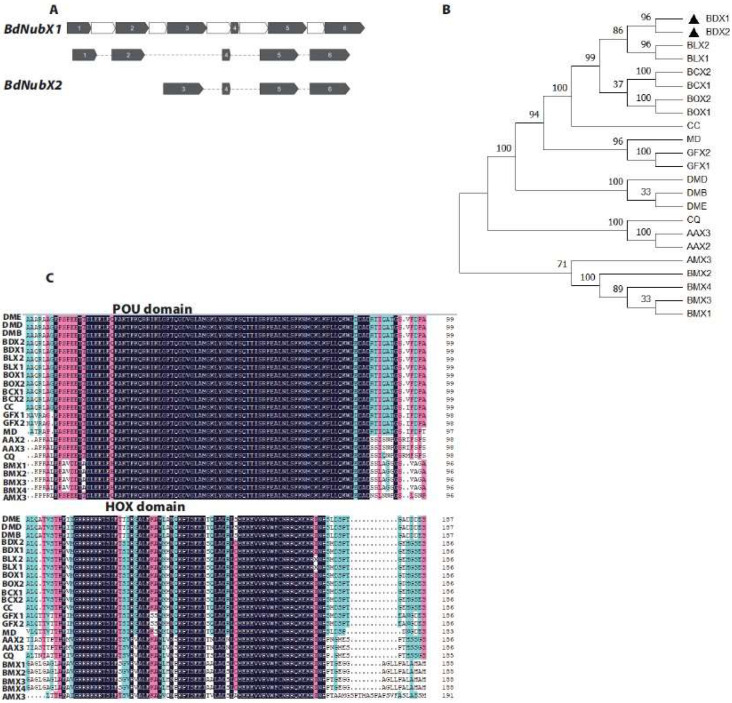
Identification of *BdNub* gene. (**A**): The alternative splicing of *BdNub*. Exons are indicated by grey color, Introns are indicated by white color. (**B**): Phylogenetic analysis of the *Nub* gene, *BdNub* were aligned with *Nub* genes from 11 other insect species. Gene accession numbers were given in parentheses. Black triangle(the location of the target gene). (**C**): Alignment of POU and HOX domain amino acid sequence. Identical sequences were shown in black. 75% conserved amino acids were shown on pink background, and 50% conserved amino acids were shown on blue background. (**B**,**C**). BDX1, *Bactrocera dorsalis nubbin X1* (XP 011207745.1); BDX2, *Bactrocera dorsalis nubbin X2* (XP 011207746.1); BLX1, *Bactrocera latifrons nubbin X1* (XP 018783925.1); BLX2, *Bactrocera latifrons nubbin X 2*(XP 018783926.1); BCX2, *Bactrocera cucurbitae nubbin X2*(XP 011178505.1) ;BCX1, *Bactrocera cucurbitae nubbin X1* (XP 011178504.1); BOX2, *Bactrocera oleae nubbin X2* (XP 014086366.1); BOX1, *Bactrocera oleae nubbin X2*; CC, *Ceratitis capitata nubbin* (XP 004530324.1); MD, *Musca domestica nubbin* (XP 019892278.1); GFX2, *Glossina fuscipes nubbin* X2 (XP 037882400.1); GFX1, *Glossina fuscipes nubbin X1* (XP 037882399.1); DMB, *Drosophila melanogaster nubbin B* (NP 001097153.1); DMD, *Drosophila melanogaster nubbin D* (NP 476659.1) ; DME, *Drosophila melanogaster nubbin E*(NP 001285876.1) ; CQ, *Culex quinquefasciatus nubbin* (XP 001844054.1); AAX3, *Aedes aegypti nubbin X3*(XP 021704008.1); AAX2, *Aedes aegypti nubbin X2* (XP 021704007.1); AMX3, *Apis mellifera nubbin X3* (XP 006558737.1); BMX2, *Bombyx mori nubbin X2* (XP 037870381.1) ; BMX4, *Bombyx mori nubbin X4*(XP 037870383.1); BMX3, *Bombyx mori nubbin X3* (XP 037870382.1); BMX1, *Bombyx mori nubbin X1*(XP 037870380.1).

**Figure 2 insects-14-00178-f002:**
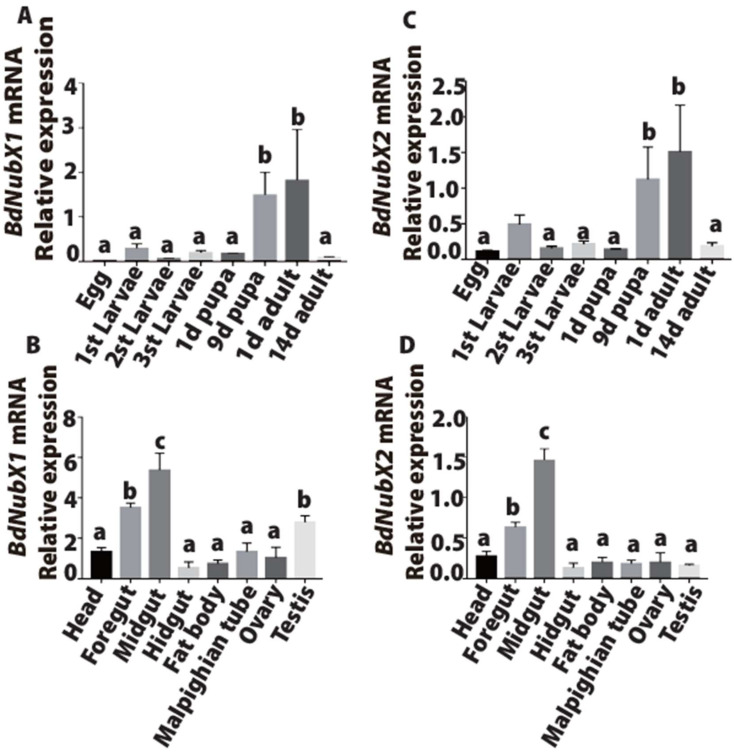
Expression profiles of *BdNubX1* and *BdNubX2.* (**A**)*: BdNubX1* Expression profile of different development stages, 20 different stage of development samples per biological replicate, three biological replicates. (**B**)*: BdNubX1* expression profile of different adult tissues, 30 different tissue samples per biological replicate, three biological replicates. (**C**)*: BdNubX2* Expression profile of different development stages, 20 different stage of development samples per biological replicate, three biological replicates. (**D**): *BdNubX2* expression profile of different tissues of adults. Different letters indicate statistically significant differences in *BdNub* isoforms expression, 30 different tissue samples per biological replicate, three biological replicates. *p* < 0.05, Tukey’s test, One way ANOVA.

**Figure 3 insects-14-00178-f003:**
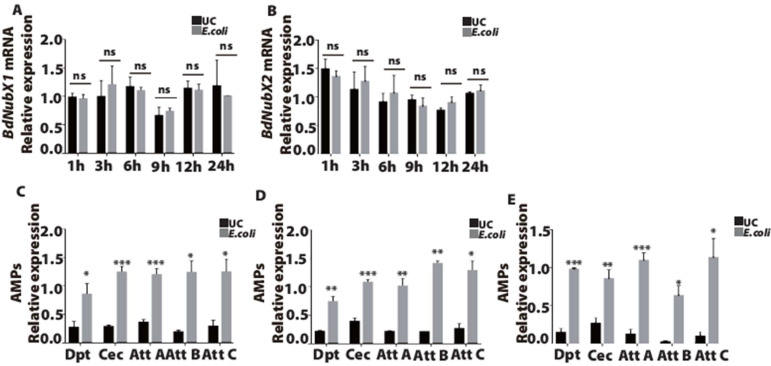
The immune response of *BdNub* after *E. coli* systemic infection. (**A**) The relative expression level of *BdNubX1* after *E. coli* systemic infection. (**B**) The relative expression level of *BdNubX2* after *E. coli* systemic infection. (**C**–**E**) The relative expression of AMPs genes at 3 h, 6 h, 12 h after *E. coli* systemic infection, respectively, (**A**–**E**) 30 flies samples per biological replicate, three biological replicates. Gene expressions were normalized to the reference gene RP49. UC: unchallenged flies. * *p* < 0.05, ** *p* < 0.01, *** *p* < 0.001, Student’s *t*-test.

**Figure 4 insects-14-00178-f004:**
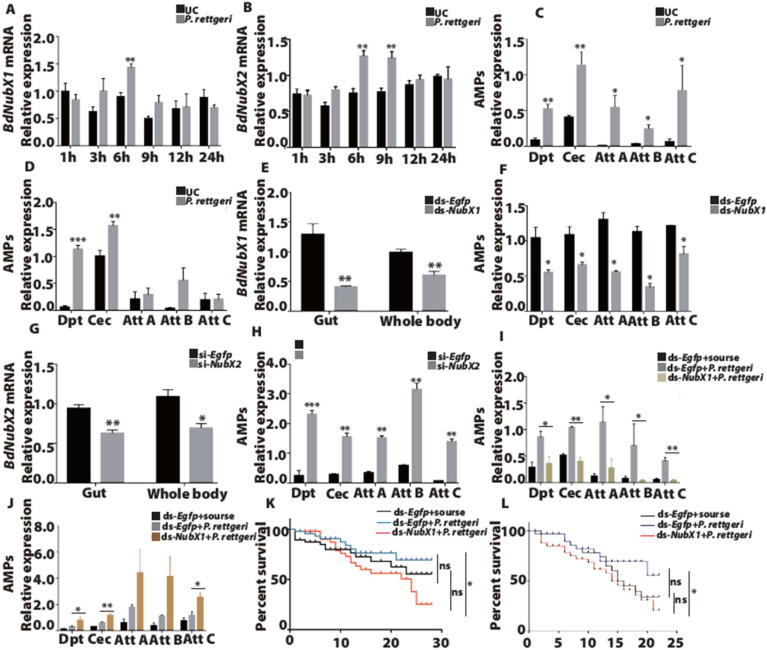
The immune response of *BdNub* after *P. rettgeri* oral infection. (**A**,**B**) The relative expression level of *BdNubX1* and *BdNubX2* after *P. rettgeri* oral infection. (**C**,**D**) The relative expression of the AMPs genes at 6h and 9h after oral infection, respectively. (**E**) The RNAi effect of *BdNubX1* dsRNA injection. (**F**) The expression levels of AMPs genes in *BdNubX1* RNAi flies. (**G**) The RNAi effect of *BdNubX1* siRNA injection. (**H**) The expression levels of AMPs genes in *BdNubX2* RNAi flies. (**I**,**J**) The AMPs genes expressions at 6 h after *P. rettgeri* oral infection in *BdNubX1* RNAi flies (I) and *BdNubX2* RNAi flies (**J**). ds*Egfp* treated group was used as the control for RNAi. (**K**,**L**) The survival of *BdNubX1* RNAi and *BdNubX2* RNAi flies after rate after *P. rettgeri* oral infection. (**A**–**L**) 30 flies samples per biological replicate, three biological replicates. * *p* < 0.05, ** *p* < 0.01, *** *p* < 0.001, Student’s *t*-test. For survival assay, ns, non-significance, Log-rank (Mantel-Cox) test.

**Figure 5 insects-14-00178-f005:**
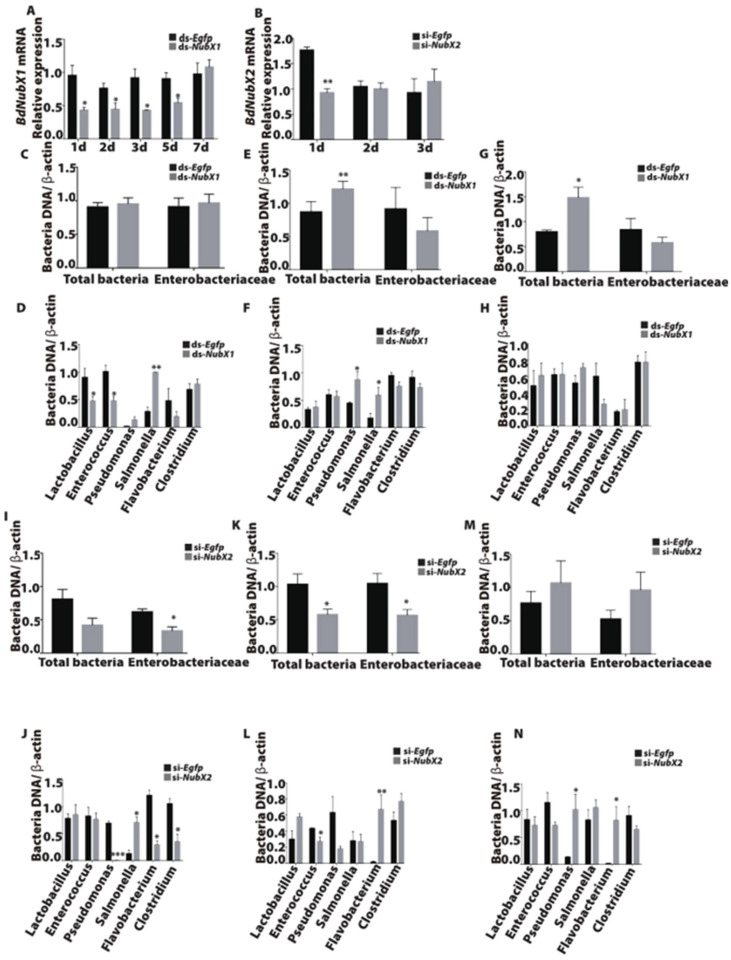
Effect of *BdNubX1* and *BdNubX2* RNAi on gut microbiota. (**A**,**B**) *BdNubX1* and *BdNubX2* expression levels after dsRNA and siRNA injection, respectively. (**C**) The gut total bacteria and *Enterobacteriaceae* abundance at 3d in *BdNubX1* RNAi flies. (**D**) The different genus bacteria abundance at 3 d in *BdNubX1* RNAi flies. (**E**) The gut total bacteria and *Enterobacteriaceae* abundance at 5d in *BdNubX1* RNAi flies. (**F**) The different genus bacteria abundance at 5 d in *BdNubX1* RNAi flies. (**G**) The gut total bacteria and *Enterobacteriaceae* abundance at 7 d in *BdNubX1* RNAi flies. (**H**) The different genus bacteria abundance at 7d in *BdNubX1* RNAi flies. (**I**) The gut total bacteria and *Enterobacteriaceae* abundance at 3 d in *BdNubX2* RNAi flies. (**J**) The different genus bacteria abundance at 3 d in *BdNubX2* RNAi flies. (**K**) The gut total bacteria and *Enterobacteriaceae* abundance at 5 d in *BdNubX2* RNAi flies. (**L**) The different genus bacteria abundance at 5 d in *BdNubX2* RNAi flies. (**M**) The gut total bacteria and *Enterobacteriaceae* abundance at 7 d in *BdNubX2* RNAi flies. (**N**) The different genus bacteria abundance at 7 d in *BdNubX2* RNAi flies, (**A**–**N**) 30 flies guts samples per biological replicate, three biological replicates. * *p* < 0.05, ** *p* <0.01, *** *p* < 0.001, Student’s *t*-test.

**Table 1 insects-14-00178-t001:** The primers used for Quantitative Real-time PCR.

Primer Name	Sequence	Target Gene	Amplicon Size (bp)
*QRpl32* F	5′-CCCGTCATATGCTGCCAACT-3′	*Rpl32*	148 bp
*QRpl32* R	5′-GCGCGCTCAACAATTTCCTT-3′		
*QBdNub*X1 F	5′-GCAGTAATGTGCCCCAGAAG-3′	*Nub*X1	115 bp
*QBdNub*X1 R	5′-AACGCAGACGTAGCGGTAAC-3′		
*QBdNub*X2 F	5′-GTCGAGCATCGAGGTGTTTT-3′	*Nub*X2	105 bp
*QBdNub*X2 R	5′-AGTGTCTGAGCGCTTGTGTG-3′		
*QBdDpt* F	5′-GCATAGATTTGAGCCTTGACACAC-3′	*Diptcin*	110 bp
*QBdDpt* R	5′-GCCATATCGTCCGCCCAAAT-3′		
*QBdCec* F	5′-GGCAAGAAAATTGAGCGGGT-3′	*Cecropin*	100 bp
*QBdCec* R	5′-CCTTCAATGTTGCTGCCACA-3′		
*QBdAttA* F	5′-GTGGCAACCTTAATTGGGCG-3′	*Attcin A*	106 bp
*QBdAttA* R	5′-AGATTGGAACTTGCGCCGTA-3′		
*QBdAttB* F	5′-ACACGCTTGGACTTGACAGG-3′	*Attcin B*	93 bp
*QBdAttB* R	5′-ATGAGTCAATCCCAAGCCGG-3′		
*QBdAttC* F	5′-GAGTTGGCCGGTAGAGCAAA-3′	*Attcin C*	104 bp
*QBdAttC* R	5′-GTAGTCGCGTTGTCCACTCA-3′		
*QBdDef* F	5′-CTGGAAAAGTCAATGGGCCG-3′	*Defensin*	105 bp
*QBdDef* R	5′-AAGCGATACAATGGACAGCG-3′		
*QBdactinF*	5′-TCGATCATGAAGTGCGATGT-3	*β* *-actin*	101 bp
*QBdactinR*	5′-ATCAGCAATACCGGGGTACA-3		

**Table 2 insects-14-00178-t002:** The primers used for synthesis of dsRNA.

*BdNub*X1 T7F	5′-GGATCCTAATACGACTCACTATAGGACCAGGCATTTTGAACCCA-3′
*BdNub*X1 T7R	5′-GGATCCTAATACGACTCACTATAGGTGATCCGCTGACTCCGTCT-3′
*Egfp* T7F	5′-GGATCCTAATACGACTCACTATAGGACGTAAACGGCCACAAGTTC-3
*Egfp* T7R	5′-GGATCCTAATACGACTCACTATAGGAAGTCGTGCTGCTTCATGTG-3′
*Si-BdNubX2*	5′-GTGGGCACATAATGCAGAA-3′

**Table 3 insects-14-00178-t003:** The primers used for different bacteria genera.

Target Bacteria	Primer Name	Sequence	Resources	Target Gene	Amplicon Size (bp)
Total bacteria	*Tol F*	5′-TCCTACGGGAGGCAGCAGT-3′	(Guo et al., 2008) [23,27]	16S RNA	466 bp
	*Tol R*	5′-GGACTACCAGGGTATCTATCCTGTT-3′			
*Enterobacteriaceae*	*Ent F*	5′-CATTGACGTTACCCGCAGAAGAAGC-3′	(Bartosch et al., 2004) [23,28]	16S RNA	195 bp
	*Ent R*	5′-CTCTACGAGACTCAAGCTTGC-3′			
*Enterococcus*	*Eco F*	5′-CCCTTATTGTTAGTTGCCATCATT-3	(Rinttilä et al., 2004) [29]	16S RNA	144 bp
	*Eco R*	5′-ACTCGTTGTACTTCCCATTGT-3′			
*Lactobacillus*	*Lac F*	5′-AGCAGTAGGGAATCTTCCA-3′	(Rinttilä et al., 2004) [29]	16S RNA	341 bp
	*Lac R*	5′-CACCGCTACACATGGAG-3′			
*Flavobacterium*	*Flavo F*	5′-ATTGGGTTTAAAGGGTCC-3′	(Abell et al., 2005) [30]	16SRNA	349 bp
	*Flavo R*	5′-CCGTCAATTCCTTTGAGTTT-3′			
*Salmonella*	*Salmon F*	5′-ACAGTGCTCGTTTACGACCTGAAT-3′	(Ahmed et al., 2008) [31]	*invA*	244 bp
	*Salmon R*	5′-AGACGACTGGTACTGATCGATAAT-3′			
*Pseudomonas*	*Pseudo F*	5′-CAAAACTACTGAGCTAGAGTACG-3′	(Matsuda et al., 2009) [32]	16S RNA	215 bp
	*Pseudo R*	5′-TAAGATCTCAAGGATCCCAACGGCT-3′			
*Clostridium*	*Cclos F*	5′-AATCTTGATTGACTGAGTGGCGGAC-3′	(Bartosch et al., 2004) [28]	16S RNA	148 bp
	*Cclos R*	5′-CCATCTCACACTACCGGAGTTTTTC-3′			

## Data Availability

The data presented in this study are available on request from the corresponding author.

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
