# Peer review of "BdNub Is Essential for Maintaining gut Immunity and Microbiome Homeostasis in Bactrocera dorsalis"

_insects, 2023, doi:10.3390/insects14020178_

Round 1
Reviewer 1 Report
In this manuscript, the authors analyzed the role of Nub gene in regulating IMD pathway of Bactrocera dorsalis. The results showed that the two transcription isoforms BdNubX1 and BdNubX2 regulate IMD pathway differently and they may involve in gut microbiota homeostasis. In general, the work is significant for understanding insect immune regulation and the relationship between gut microbiota and host immune system.
Major points:
1. In general, the writing is clear and easy to understand. However, there are many mistakes throughout the text. Please check the writing again in detail. Some examples:
(1) Line 55, 63……Latin names should be italic.
(2) Line 27, 28……The NUB gene → Nub gene.
(3) Line 13: No 。in English.
(4) Line 13: Bactrocera dorsalis to study its……
(5) Line 16, 78, 79…… There should be a space between two sentences.
(6) Line 16, 20: bacteria → bacterium
(7) Line 21: Providencia rettgeri infection,
(8) Mistakes in spelling:
Line 87: Larva
Line 187: different bacterial genera (family)
Line 278: inducer
……
(9) Please list the references in Table 3.
2. Methods:
(1) Line165, 171:Bacterial infection: To provide the dose of infection is better than concentration.
(2) Quantification of bacterial groups. qPCR is a good method to quantify the relative abundance of different bacterial groups. Since the primers come from old references, some may not be suitable any more. It is better to present the bacterial community structure changes with a second method, e.g. 16S rDNA amplicon analysis.
Reviewer 3 Report
Summary: The manuscript “BdNub is essential for maintaining gut immunity and microbiome homeostasis in Bactrocera dorsalis” by Gu and Zhang et al investigated role the of NUB gene in regulating host microbiota in horticultural pest Bactrocera dorsalis. Based on NCBI database search, the authors found 2 isoforms of the Nub gene in B. dorsalis, BdNubX1 and BdNubX2. They initially identified the temporal and spatial expression of BdNub gene by analyzing different life stages and tissues of B. dorsalis. Subsequent molecular assays showed a significant upregulation of BdNubX1, BdNubX2, and antimicrobial peptides (AMPs) after gut infection. The regulatory functions of BdNubX1 and BdNubX2 were measured by RNAi assays. Further DNA-based assays revealed that BdNubX1 and BdNubX2 are associated with gut microbiota composition. The authors concluded that the Nub gene contributes to maintaining gut microbiota homeostasis in Bactrocera dorsalis. The authors have conducted a systematic analysis of BdNub genes and microbiota to test their hypothesis. Overall, the data is very interesting, and the manuscript is well written, but the method section needs to be improved. I have a few comments for the authors to consider.
Major Comment:
One important comment I have is using dsEgfp / siEgfp treatment only as a control and not including untreated control in RNAi experiments. dsEgfp / siEgfp injection might also trigger changes in microbiome composition and structure. I think including “untreated control” treatment would have immensely benefited in RNAi assays (survival rate) and provided a clear view of dynamics in microbial composition. The samples could even be used for absolute quantification of specific microbes, if opted. To address this, please specify any specific reasons for choosing dsEgfp / siEgfp treatment only as a control in this study.
Other comments:
Line 101, 103: Add Country name along with University or Province name.
Line 109: “Samples” Which samples do these refer to? Please add the details.
Line 125-132: If these are the same samples that are being referred to in Line 109, I suggest moving this section before RNA extraction and cDNA synthesis.
Sections 2.4, 2.6, 2.7, 2.8, and 2.9: No information was given regarding the replicate number for any experiment. Please add these details in the text and include the replicate number in the respective figure legends. How many flies/tissues constitute a replicate? If the number varies for each assay, please provide the relevant information in the respective section.
Line 146: dsRNA synthesis and RNA interference: This section is key to this paper, and I suggest the authors provide a detailed description of this section. What age flies were chosen for RNA interference experiments? Is flies age consistent across all experiments?
Line 276: Does “previous data” mean is it unpublished data or a published study? If it is published, please provide the correct reference.
Based on lines 168-169, P. rettgeri was isolated from the B. dorsalis gut, yet it triggered changes in gut composition. Is this bacterium a common gut bacteria of B. dorslis? Would it be true with any other gut symbiont, for example, Psuedomonas? Would these experiment benefits by using sterile insects to understand the IMD response to the colonization of gut symbionts?
Table 1: Please add new columns for “Target gene” and “Amplicon size” for each primer set.
Table 3: Host β-actin primers are missing. Also please add new columns for “Target gene” and “Amplicon Size” for each primer size?
Figure 1B and 1C: Font size is too small to read. Please increase the size of the images for readability.
Figure 5: As presented, I find this figure a bit hard to follow. Please consider rearranging figures by gene and by day if possible. For example, Figure 5C, 5E and 5G be placed in one row and the row below will have 5D, 5F, 5H representing 3d, 5d and 7d respectively. The same pattern can be followed for si-NubX2 figures as well.
Round 2
Reviewer 1 Report
The manuscript has been improved. Please make revision in the following places.
Line 90: study
Line 135/137: isolated from
Line 220/227: per biological replicate
Line 237: Please check if this sentence is complete or not.
Line 250: Nub gene
Line 273/285: All Latin names should be italics.
Author Response
Thank you very much for your comments. We have corrected them accordingly.
Reviewer 2 Report
The authors have answered properly to majority of the questions and accept the majority of the suggestions. However, two points remained unsolved, therefore I strongly suggest that authors revise those parts of the manuscript:
Introduction
Although authors added a brief description of the methods used to achieve the aims of the study, they still comment on the results which is redundant. Instead, it would be much better if the authors explain what was the purpose of the study and how this study can help to better understand the life cycle of how this major horticultural and agricultural pest and consequently control its invasion.
Results
Please avoid discussing on your results in the Result section and leave it for Discussion section. In that way you will also avoid unnecessary repetition and make results clearer. – Authors did not address this point.
Author Response
Thank you for your comments. We have revised the manuscript accordingly.